# SVFT: Parameter-Efficient Fine-Tuning with Singular Vectors

**Vijay Lingam** [* 1 2]   **Atula Tejaswi** [* 1]   **Aditya Vavre** [* 1]   **Aneesh Shetty** [* 1]   **Gautham Krishna Gudur** [* 1]
**Joydeep Ghosh** [1]   **Alex Dimakis** [1]   **Eunsol Choi** [1]   **Aleksandar Bojchevski** [3]   **Sujay Sanghavi** [1]

## Abstract

Popular parameter-efficient fine-tuning (PEFT) methods, such as LoRA and its variants, freeze pre-trained model weights $\mathbf{W}$ and inject learnable matrices $\mathbf{\Delta W}$. These $\mathbf{\Delta W}$ matrices are structured for efficient parameterization, often using techniques like low-rank approximations or scaling vectors. However, these methods typically show a performance gap compared to full fine-tuning. Although recent PEFT methods have narrowed this gap, they do so at the cost of additional learnable parameters. We propose SVFT, a *simple* approach that fundamentally differs from existing methods: the structure imposed on $\mathbf{\Delta W}$ depends on the specific weight matrix $\mathbf{W}$. Specifically, SVFT updates $\mathbf{W}$ as a sparse combination of outer products of its singular vectors, training only the coefficients (scales) of these sparse combinations. This approach allows fine-grained control over expressivity through the number of coefficients. Extensive experiments on language and vision benchmarks show that SVFT recovers up to **96%** of full fine-tuning performance while training only **0.006 to 0.25%** of parameters, outperforming existing methods that only recover up to **85%** performance using **0.03 to 0.8%** of the trainable parameter budget.

## 1. Introduction

Large-scale foundation models are often adapted for specific downstream tasks after pre-training. Parameter-efficient fine-tuning (PEFT) facilitates this adaptation efficiently by learning a minimal set of new parameters, thus creating an "expert" model. For instance, Large Language Models (LLMs) pre-trained on vast training corpora are fine-tuned for specialized tasks such as text summarization (Hermann et al., 2015; Zhang et al., 2020), sentiment analysis (Raffel et al., 2020; Liu et al., 2019), and code completion (Rozière et al., 2024) using instruction fine-tuning datasets. Although full fine-tuning (Full-FT) is a viable method to achieve this, it requires re-training and storing all model weights, making it impractical for deployment with large foundation models.

To address these challenges, PEFT techniques (Houlsby et al., 2019) (e.g., LoRA (Hu et al., 2022)) were introduced to significantly reduce the number of learnable parameters compared to Full-FT, though often at the cost of performance. DoRA (Liu et al., 2024a) bridges this performance gap by adding more learnable parameters and being more expressive than LoRA. Almost all these methods apply a low-rank update additively to the frozen pre-trained weights, potentially limiting their expressivity. Furthermore, these adapters are agnostic to the structure and geometry of the weight matrices they modify. Finally, more expressive PEFT methods (e.g., LoRA, DoRA, BOFT (Liu et al., 2024b)) still accumulate a considerable portion of learnable parameters even in their most efficient configuration (e.g., setting rank=1 in LoRA and DoRA). The storage requirements for the learnable adapters can grow very quickly when adapting to a large number of downstream tasks (Kopiczko et al., 2024).

In this work we address the following research question: *Is it possible to narrow the performance gap between PEFT and Full-FT, while being highly parameter-efficient?* Towards this end, we propose SVFT: Singular Vectors guided Fine-Tuning — a *simple* approach that involves updating an existing weight matrix by adding to it a sparse weighted combination of *its own singular vectors*. The structure of the induced perturbation in SVFT depends on the specific matrix being perturbed, setting it apart from all previous approaches. Our contributions can be summarized as follows:

- We introduce SVFT, a new PEFT method. Given a weight matrix $\boldsymbol{W}$, SVFT involves adapting it with a matrix $\Delta \boldsymbol{W} := \sum_{(i,j) \in \Omega} m_{ij} \boldsymbol{u}_i \boldsymbol{v}_j^T$, where $\{\boldsymbol{u}_i\}$ and $\{\boldsymbol{v}_j\}$ are the left and right singular vectors of $\boldsymbol{W}$, $\Omega$ is an a-priori fixed sparsity pattern, and $m_{ij}$ for $(i,j) \in \Omega$ are learnable parameters. By controlling $|\Omega|$ we can efficiently explore the accuracy vs parameters trade-off.

---
[*]Equal contribution   [1]University of Texas at Austin   [2]CISPA Helmholtz Center for Information Security   [3]University of Cologne. Correspondence to: Vijay Lingam <vijaylingam0810@gmail.com>.

Accepted to the Workshop on Advancing Neural Network Training at International Conference on Machine Learning (WANT@ICML 2024).

- SVFT achieves higher downstream accuracy, as a function of the number of trainable parameters, as compared to several popular PEFT methods (see Figure 1) and over several downstream tasks across both vision and language tasks. Our method recovers up to **96%** of full fine-tuning performance while training only **0.006 to 0.25%** of parameters, outperforming existing methods that only recover up to **85%** performance using **0.03 to 0.8%** of the trainable parameter budget.

We introduce four variants for parameterizing weight updates, namely: *Plain*, *Random*, *Banded*, and *Top-k* in SVFT (which differ in their choices of the fixed sparsity pattern $\Omega$) and validate these design choices empirically. Additionally, we theoretically show that for any fixed parameters budget, SVFT can induce a higher rank perturbation compared to previous PEFT techniques.

## 2. Related Work

Recent advancements in large language models (LLMs) have emphasized the development of PEFT techniques to enhance the adaptability and efficiency of large pre-trained language models.

**LoRA.** A notable contribution in this field is Low-Rank Adaptation (LoRA) (Hu et al., 2022), which freezes the weights of pre-trained models and integrates trainable low-rank matrices into each transformer layer. For a pre-trained weight matrix $\boldsymbol{W}_0 \in \mathbb{R}^{d \times n}$, LoRA constraints the weight update $\Delta \boldsymbol{W}$ to a low-rank decomposition: $\boldsymbol{h} = \boldsymbol{W}_0 \boldsymbol{x} + \Delta \boldsymbol{W} \boldsymbol{x} = \boldsymbol{W}_0 \boldsymbol{x} + \underline{\boldsymbol{B} \boldsymbol{A}} \boldsymbol{x}$, where $\boldsymbol{B} \in \mathbb{R}^{d \times r}$, $\boldsymbol{A} \in \mathbb{R}^{r \times n}$ and rank $r \ll \min(d, n)$. We underline the (trainable) parameters that are updated via gradient descent.

**LoRA variants.** We highlight some recent approaches that further improve the vanilla LoRA architecture. Vector-based Random Matrix Adaptation (VeRA) (Kopiczko et al., 2024) minimizes the number of trainable parameters by utilizing a pair of low-rank random matrices shared between layers and learning compact scaling vectors while maintaining performance comparable to LoRA. Formally, VeRA can be expressed as: $\boldsymbol{h} = \boldsymbol{W}_0 \boldsymbol{x} + \Delta \boldsymbol{W} \boldsymbol{x} = \boldsymbol{W}_0 \boldsymbol{x} + \underline{\boldsymbol{\Lambda}_b} \boldsymbol{B} \underline{\boldsymbol{\Lambda}_d} \boldsymbol{A} \boldsymbol{x}$, where $\boldsymbol{A}$ and $\boldsymbol{B}$ are initialized randomly, frozen, and shared across layers, while $\boldsymbol{\Lambda}_b$ and $\boldsymbol{\Lambda}_d$ are trainable diagonal matrices.

An alternative approach, Weight-Decomposed Low-Rank Adaptation (DoRA) (Liu et al., 2024a), decomposes pre-trained weight matrices into magnitude and direction components, and applies low-rank updates for directional updates, reducing trainable parameters and enhancing learning capacity and training stability. DoRA can be expressed as: $\boldsymbol{h} = \underline{\boldsymbol{m}} \frac{\boldsymbol{W}_0 + \Delta \boldsymbol{W}}{\|\boldsymbol{W}_0 + \Delta \boldsymbol{W}\|_c} \boldsymbol{x} = \underline{\boldsymbol{m}} \frac{\boldsymbol{W}_0 + \underline{\boldsymbol{B} \boldsymbol{A}}}{\|\boldsymbol{W}_0 + \underline{\boldsymbol{B} \boldsymbol{A}}\|_c} \boldsymbol{x}$, where $\| \cdot \|_c$ denotes the vector-wise norm of a matrix

across each column. Similar to LoRA, $\boldsymbol{W}_0$ remains frozen, whereas the magnitude vector $\boldsymbol{m}$ (initialized to $\|\boldsymbol{W}_0\|_c$) and low-rank matrices $\boldsymbol{A}, \boldsymbol{B}$ contain trainable parameters.

AdaLoRA (Zhang et al., 2023) adaptively distributes the parameter budget across weight matrices based on their importance scores and modulates the rank of incremental matrices to manage this allocation effectively. PiSSA (Principal Singular Values and Singular Vectors Adaptation) (Meng et al., 2024) is another variant of LoRA, where matrices $\boldsymbol{A}, \boldsymbol{B}$ are initialized with principal components of SVD and the remaining components are used to initialize $\boldsymbol{W}_0$. FLoRA (Wen & Chaudhuri, 2024) enhances LoRA by enabling each example in a mini-batch to utilize distinct low-rank weights, preserving expressive power and facilitating efficient batching, thereby extending the domain adaptation benefits of LoRA without batching limitations.

**Other PEFT variants.** Orthogonal Fine-tuning (OFT) (Qiu et al., 2023) modifies pre-trained weight matrices through orthogonal reparameterization to preserve essential information. However, it still requires a considerable number of trainable parameters due to the high dimensionality of these matrices. Butterfly Orthogonal Fine-tuning (BOFT) (Liu et al., 2024b) extends OFT's methodology by incorporating Butterfly factorization thereby positioning OFT as a special case of BOFT. Unlike the additive low-rank weight updates utilized in LoRA, BOFT applies multiplicative orthogonal weight updates, marking a significant divergence in the approach but claims to improve parameter efficiency and fine-tuning flexibility. BOFT can be formally expressed as: $\boldsymbol{h} = (\underline{\boldsymbol{R}(m, b)} \cdot \boldsymbol{W}_0)\boldsymbol{x}$, where the orthogonal matrix $\boldsymbol{R}(m, b) \in \mathbb{R}^{d \times d}$ is composed of a product of multiple orthogonal butterfly components. When $m = 1$, BOFT reduces to block-diagonal OFT with block size $b$. When $m = 1$ and $b = d$, BOFT reduces to the original OFT with an unconstrained full orthogonal matrix.

## 3. Method

In this section, we introduce Singular Vectors guided Fine-Tuning (SVFT). The main innovation in SVFT lies in applying structure/geometry-aware weight updates.

### 3.1. SVFT Formulation

We now formally describe our method, SVFT for parameter-efficient fine-tuning of a pre-trained model. Let $\boldsymbol{W}_0 \in \mathbb{R}^{d_1 \times d_2}$ denote a weight matrix in the pre-trained model. For instance, in a transformer block, this could be the key matrix, the query matrix, a matrix in the MLP, etc. We add a structured, learned $\Delta \boldsymbol{W}$ to this matrix as follows.

As a first step, we compute the Singular Value Decomposition (SVD) of the given matrix: $\boldsymbol{W}_0 = \boldsymbol{U} \boldsymbol{\Sigma} \boldsymbol{V}^T$. That is, $\boldsymbol{U}$ is the $d_1 \times d_1$ matrix of left singular vectors (i.e., its

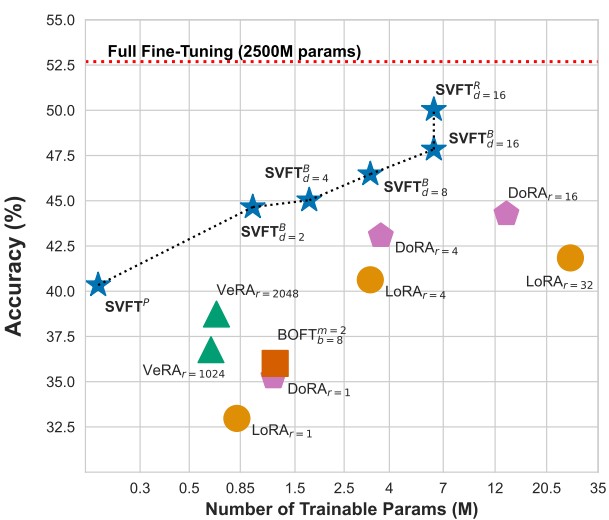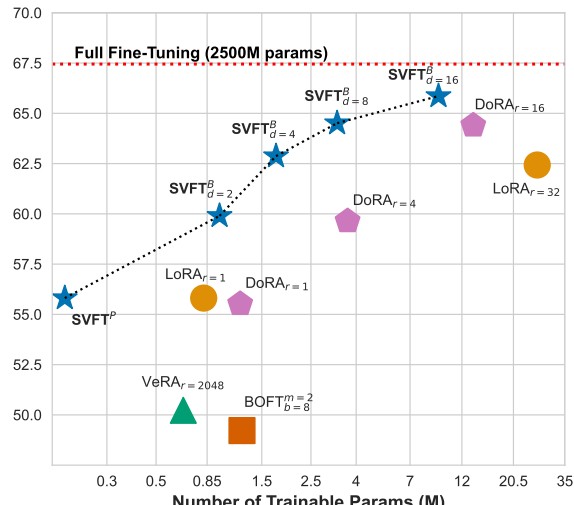

*Figure 1.* Performance vs total trainable parameters for GSM-8K (left) and Commonsense Reasoning (right) on Gemma-2B. $\text{SVFT}_{d=16}^{B/R}$ outperforms $\text{DoRA}_{r=8/16}$ with 75% less trainable parameters.

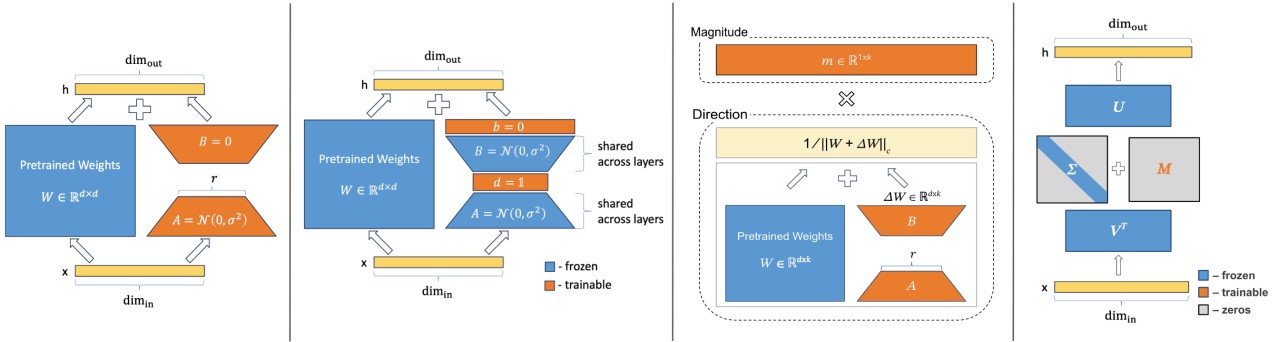

*Figure 2.* Schematic comparison of LoRA, VeRA, DoRA, and SVFT (left to right).

columns are orthonormal), $\boldsymbol{V}^T$ is the $d_2 \times d_2$ matrix of right singular vectors (i.e., its rows are orthonormal), and $\boldsymbol{\Sigma}$ is a $d_1 \times d_2$ diagonal matrix. Then, we parameterize our weight update as $\Delta \boldsymbol{W} = \boldsymbol{U} \underline{\boldsymbol{M}} \boldsymbol{V}^T$, where $\boldsymbol{U}, \boldsymbol{V}$ are fixed and frozen, while $\underline{\boldsymbol{M}}$ is a $d_1 \times d_2$ **sparse trainable matrix with pre-determined and fixed sparsity pattern**[1]. That is, we first pre-determine a small fixed set of elements in $\boldsymbol{M}$ that will be allowed to be non-zero and train only those elements. The forward pass for SVFT can be written as,

$$\boldsymbol{h} = \boldsymbol{W}_0 x + \Delta \boldsymbol{W} x = \boldsymbol{U}(\boldsymbol{\Sigma} + \underline{\boldsymbol{M}})\boldsymbol{V}^T \boldsymbol{x} \qquad (1)$$

We explore four choices for $\Omega$, the a-priori fixed sparsity pattern of $\underline{\boldsymbol{M}}$.

**Plain** $(\text{SVFT}^P)$. In this variant, we constrain $\underline{\boldsymbol{M}}$ to be a diagonal matrix, which can be interpreted as adapting singular values and reweighting the frozen singular vectors. Since only the diagonal elements are learned, this is the

most parameter-efficient SVFT variant.

**Banded** $(\text{SVFT}_d^B)$. In this approach, we populate $\underline{\boldsymbol{M}}$ using a banded matrix, progressively making off-diagonals learnable. Specifically, for constants $z_1$ and $z_2$, $\underline{\boldsymbol{M}}_{ij} = 0$ if $j < i - z_1$ or $j > i + z_2$, where $z_1, z_2 \geq 0$. In our experiments, we set $z_1 = z_2 = d$ to induce off-diagonal elements that capture additional interactions beyond those represented by singular values. This banded perturbation induces local interactions, allowing specific singular values to interact with their immediate neighbors, ensuring smoother transitions. This method, although deviating from the canonical form of SVD, provides a mechanism to capture localized interactions.

**Random** $(\text{SVFT}_d^R)$. A straightforward heuristic for populating $\underline{\boldsymbol{M}}$ involves randomly selecting $k$ random elements to be learnable.

**Top-$k$** $(\text{SVFT}_d^T)$. The final design choice we explore involves computing the alignment between the left and right singular vectors as $\boldsymbol{u}_i^T \boldsymbol{v}_j$. We then select the top-$k$ elements and make them learnable. However, note that this

---

[1] Learnable parameters are underlined.

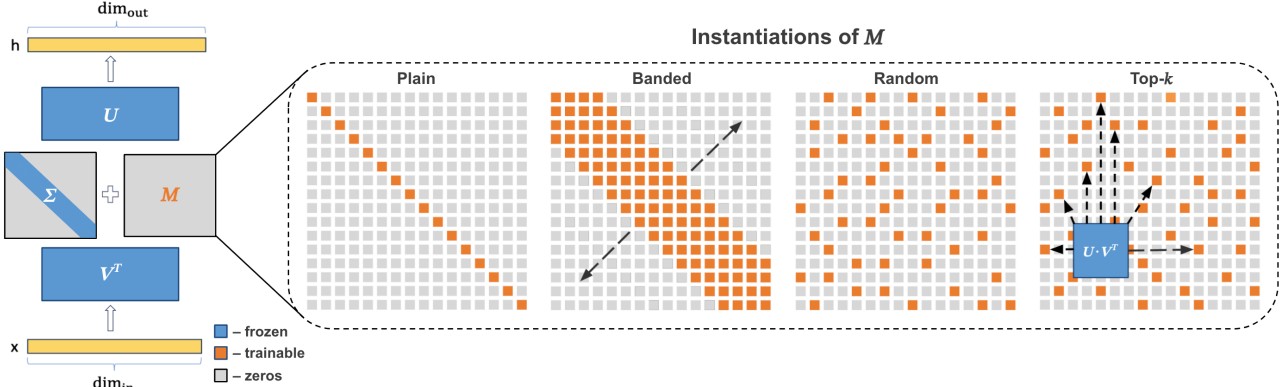

*Figure 3.* An Overview of SVFT. The original weights $\boldsymbol{W}$ are decomposed into $\boldsymbol{U}, \boldsymbol{\Sigma}, \boldsymbol{V}$. We introduce the following parameterizations for $\boldsymbol{M}$: Plain, Banded, Random, and Top-$k$. We highlight trainable parameters in orange.

only works when left/right singular vectors have the same size. A possible interpretation of this is we make only the top-$k$ strong interactions between singular vector directions learnable.

We illustrate these SVFT design choices in Figure 3. Our empirical results demonstrate that these simple design choices significantly enhance performance compared to state-of-the-art PEFT methods. Note that SVFT$^P$ has a fixed number of learnable parameters, while the remaining variants are configurable. We hypothesize that further innovation is likely achievable through optimizing the sparsity pattern of $\underline{\boldsymbol{M}}$, including efficient learned-sparsity methods. In this paper, we explore these four choices to validate the overall idea: determining a perturbation using the singular vectors of the matrix that is being perturbed.

### 3.2. Properties of SVFT

We highlight some properties of SVFT in the following lemma and provide insights into how its specific algebraic structure compares and contrasts with baseline methods.

**Lemma:** Let $\boldsymbol{W}_0$ be a matrix of size $d_1 \times d_2$ with SVD given by $\boldsymbol{U\Sigma V}^T$. Consider an updated final matrix $\boldsymbol{W}_0 + \boldsymbol{UMV}^T$, where $\boldsymbol{M}$ is a matrix of the same size as $\boldsymbol{\Sigma}$, which may or may not be diagonal. Then, the following holds:

*(a) Structure:* If $\boldsymbol{M}$ is also diagonal (i.e., the plain SVFT), then the final matrix $\boldsymbol{W}_0 + \boldsymbol{UMV}^T$ has $\boldsymbol{U}$ as its left singular vectors and $\text{sign}(\boldsymbol{\Sigma} + \boldsymbol{M})\boldsymbol{V}^T$ as its right singular vectors. That is, its singular vectors are unchanged, except for possible sign flips. Conversely, if $\boldsymbol{M}$ is *not* diagonal (i.e., variants of SVFT other than plain), then $\boldsymbol{U}$ and $\boldsymbol{V}$ may no longer be the singular directions of the final matrix $\boldsymbol{W}_0 + \boldsymbol{UMV}^T$.

*(b) Expressivity:* Given *any* target matrix $\boldsymbol{P}$ of size $d_1 \times d_2$, there exists an $\boldsymbol{M}$ such that $\boldsymbol{P} = \boldsymbol{W}_0 + \boldsymbol{UMV}^T$. That is, if $\boldsymbol{M}$ is fully trainable, any target matrix can be realized using this method.

*(c) Rank:* If $\boldsymbol{M}$ has $k$ non-zero elements, then the rank of the update $\boldsymbol{UMV}^T$ is at most $\min\{k, \min\{d_1, d_2\}\}$. For the same number of trainable parameters, SVFT can produce a much higher rank perturbation than LoRA (eventually becoming full rank), but in a constrained structured subspace.

We provide our proofs in Appendix A. Building on this lemma, we now compare the form of the SVFT update with LoRA and VeRA. SVFT's $\Delta \boldsymbol{W}$ can be written as a sum of rank-one matrices:

$$\Delta \boldsymbol{W} = \sum_{(i,j) \in \Omega} \underline{m_{ij}} \boldsymbol{u}_i \boldsymbol{v}_j^T \tag{2}$$

where $\boldsymbol{u}_i$ is the $i^{th}$ left singular vector, $\boldsymbol{v}_j$ is the $j^{th}$ right singular vector, and $\Omega$ is the set of non-zero elements in $\boldsymbol{M}$. Thus, our method involves adding a weighted combination of specific rank-one perturbations of the form $\boldsymbol{u}_i \boldsymbol{v}_j^T$.

LoRA and VeRA updates can also be expressed as sums of rank-one matrices.

$$\Delta \boldsymbol{W}_{\text{LoRA}} = \sum_{i=1}^{r} \underline{\boldsymbol{a}_i} \underline{\boldsymbol{b}_i}^T, \quad \Delta \boldsymbol{W}_{\text{VeRA}} = \sum_{i=1}^{r} \underline{\alpha_i}(\hat{\boldsymbol{a}}_i \odot \underline{\boldsymbol{\beta}})\hat{\boldsymbol{b}}_i^T \tag{3}$$

where $\underline{\boldsymbol{a}_i}$ and $\underline{\boldsymbol{b}_j}$ are the trainable columns of $\boldsymbol{A}$ and $\boldsymbol{B}$ matrices in LoRA. In VeRA, $\hat{\boldsymbol{a}}_i$ and $\hat{\boldsymbol{b}}_i$ are random and fixed vectors, while $\underline{\boldsymbol{\alpha}}$ and $\underline{\boldsymbol{\beta}}$ represent the diagonal elements of $\boldsymbol{\Lambda}_d$ and $\boldsymbol{\Lambda}_b$ respectively.

Note that LoRA requires $d_1 + d_2$ trainable parameters per rank-one matrix, while SVFT and VeRA require only one.

Although LoRA can potentially capture directions different from those achievable by the fixed $\{\boldsymbol{u}_i, \boldsymbol{v}_j^T\}$ pairs, each of these directions incurs a significantly higher parameter cost.

VeRA captures new directions at a parameter cost similar to SVFT; however, there is a key distinction: in VeRA, each vector $\hat{\boldsymbol{a}}_i$ or $\hat{\boldsymbol{b}}_i$ appears in only one of the rank-one matrices. In contrast, in SVFT, the same vector $\boldsymbol{u}_i$ can appear in multiple terms in the summation, depending on the sparsity pattern of $M$. This results in an important difference: unlike SVFT, VeRA is *not universally expressive* – it cannot represent any target matrix $\boldsymbol{P}$. Moreover, $\hat{\boldsymbol{a}}_i, \hat{\boldsymbol{b}}_i$ are random, while $\boldsymbol{u}_i, \boldsymbol{v}_j$ depend on $\boldsymbol{W}_0$.

**Note.** SVFT requires storing both left and right singular vectors due to its computation of the SVD on pre-trained weights. While this increases memory usage compared to LoRA (which is roughly double), it remains lower than BOFT. We partially address this through system-level optimizations like mixed-precision weights (e.g., bfloat16). Further exploration of memory-reduction techniques, such as quantization, is planned as future work. Importantly, inference time and memory consumption remain the same across all methods, including SVFT, as the weights can be fused.

# 4. Experiments

## 4.1. Base Models

We adapt widely-used language models, encoder-only model (DeBERTaV3$_{base}$ (He et al., 2023)) and two decoder-only models (Gemma-2B/7B (Team et al., 2024), LLaMA-3-8B (AI, 2024)). We also experiment with vision transformer models (ViT-B/16 and ViT-L/16) (Dosovitskiy et al., 2021) pre-trained on ImageNet-21k (Deng et al., 2009), following prior work (Kopiczko et al., 2024). The complete details of our experimental setup and hyper-params configurations are provided in Appendix C.

The baselines we compare against are described below.
**Full Fine-Tuning (FT)** is the conventional method for adapting a pre-trained model to a downstream task. Here, the model is initialized with pre-trained weights, and the optimizer updates learnable parameters in all layers.
**LoRA** (Hu et al., 2022) adds trainable pairs of rank decomposition matrices in parallel to existing weight matrices. To ensure a fair comparison against LoRA, we introduce another instance of LoRA in our experiments where we adjust the rank to match the scale of trainable parameters in SVFT.
**DoRA** (Liu et al., 2024a) decomposes the original weights into magnitude and direction components for fine-tuning with LoRA. While DoRA offers competitive performance, the number of learnable parameters is more than LoRA for a given rank.
**BOFT** (Liu et al., 2024b) applies orthogonal parameterization using butterfly structures, as a generalization of

OFT (Qiu et al., 2023) where neurons are transformed with orthogonal matrices. One drawback of BOFT is that it is approximately three times slower than LoRA.
**VeRA** (Kopiczko et al., 2024) learns scaling vectors to adapt a pair of frozen random matrices shared between layers. However, these shareable matrices can become a limiting factor for models with non-uniform internal dimensions. For instance, on LLaMA-3 models, VeRA can only be applied to either $\{\boldsymbol{Q}, \boldsymbol{O}\}$ or $\{$Gate $(\boldsymbol{G})$, Up $(\boldsymbol{U})$ projection$\}$ matrices.

## 4.2. Datasets

**Language.** For natural language generation (NLG) tasks, we evaluate on GSM-8K (Cobbe et al., 2021) and MATH (Hendrycks et al., 2021) by fine-tuning on MetaMathQA-40K (Yu et al., 2023), following (Liu et al., 2024b). We also evaluate on 8 commonsense reasoning benchmarks (BoolQ (Clark et al., 2019), PIQA (Bisk et al., 2020), SIQA (Sap et al., 2019), HellaSwag (Zellers et al., 2019), Winogrande (Sakaguchi et al., 2019), ARC-easy/challenge (Clark et al., 2018), and OpenBookQA (Mihaylov et al., 2018)). We follow the setting outlined in prior work (Liu et al., 2024a; Hu et al., 2023), where the training sets of all benchmarks are amalgamated for fine-tuning. We fine-tune on 15K examples from this training set. For natural language understanding (NLU), we evaluate on the General Language Understanding Evaluation (GLUE) benchmark consisting of classification and regression tasks, in line with (Kopiczko et al., 2024; Hu et al., 2022).
**Vision.** Our experiments on vision tasks consist of 4 benchmarks: CIFAR-100 (Krizhevsky et al., 2009), Food101 (Bossard et al., 2014), RESISC45 (Ullah et al., 2022), and Flowers102 (Nilsback & Zisserman, 2008). We follow the setup from (Kopiczko et al., 2024), and fine-tune on a subset comprising 10 samples from each class.

# 5. Results

## 5.1. Performance on Language Tasks

**Natural Language Generation.** We present results on mathematical question answering against baseline PEFT techniques across three base models – varying from 2B to 8B parameters in Table 1. To ensure a comprehensive comparison, we test baseline techniques (LoRA, DoRA) with different configurations, and varying hyper-parameters like rank to cover a range of learnable parameters from low to high. Note that even when the rank is as low as 1, both methods yield more trainable parameters than SVFT$^P$. SVFT$^P$ ($\sim$0.2M) shows as much as $18\%$ relative improvement over techniques that use $6\times$ more trainable parameters (BOFT$_{m=2}^{b=8}$, LoRA$_{r=1}$). Against techniques of comparable sizes (VeRA), SVFT$^P$ achieves **15.5%** relative improvement on average. Even in the default regime, SVFT$_d^R$

*Table 1.* Performance (Accuracy) on Mathematical Reasoning (GSM-8K and MATH). #Params denote the number of trainable parameters. **bold** and underline represent best and second best performing PEFT method, respectively. SVFT offers superior/competitive performance at much lower #Params. For $\text{SVFT}_d^R$, we set $d = 16$ for Gemma and $d = 12$ for LLaMA-3 models.

| Method | Gemma-2B | | | Gemma-7B | | | LLaMA-3-8B | | |
|---|---|---|---|---|---|---|---|---|---|
| | **#Params** | **GSM-8K** | **MATH** | **#Params** | **GSM-8K** | **MATH** | **#Params** | **GSM-8K** | **MATH** |
| Full-FT | 2.5B | 52.69 | 17.94 | 8.5B | 74.67 | 25.70 | 8.0B | 64.13 | 16.24 |
| LoRA$_{r=32}$ | 26.2M | 43.06 | 15.50 | 68.8M | 76.57 | 29.34 | 56.6M | **75.89** | **24.74** |
| DoRA$_{r=16}$ | 13.5M | 44.27 | **16.18** | 35.5M | 74.52 | 29.84 | 29.1M | 75.66 | **24.72** |
| BOFT$_{m=2}^{b=8}$ | 1.22M | 36.01 | 12.13 | 2.90M | 71.79 | 28.98 | 4.35M | 67.09 | 21.64 |
| DoRA$_{r=1}$ | 1.19M | 35.25 | 13.04 | 3.26M | 74.37 | 26.28 | 2.55M | 68.30 | 21.96 |
| LoRA$_{r=1}$ | 0.82M | 32.97 | 13.04 | 0.82M | 72.4 | 26.28 | 1.77M | 68.84 | 20.94 |
| VeRA$_{r=1024}$ | 0.63M | 36.77 | 14.12 | 0.43M | 71.11 | 27.04 | 0.98M | 63.76 | 20.28 |
| SVFT$^P$ | 0.19M | 40.34 | 14.38 | 0.43M | 73.50 | 27.30 | 0.48M | 69.22 | 20.44 |
| SVFT$_d^R$ | 6.35M | **50.03** | 15.56 | 19.8M | **76.81** | **29.98** | 13.1M | **75.90** | 24.22 |

*Table 2.* Evaluation results on eight commonsense reasoning benchmarks with Gemma-7B. We follow (Liu et al., 2024a) for hyperparameter configurations, and report accuracy for all tasks. HS and WG denote HellaSwag (Zellers et al., 2019) and WinoGrande (Sakaguchi et al., 2019), respectively. SVFT$^P$ offers competitive performance at a fraction of #Params. SVFT$_{d=8}^B$ can match LoRA$_{r=32}$ with ∼7x fewer parameters.

| Method | #Params | BoolQ | PIQA | SIQA | HS | WG | ARC-e | ARC-c | OBQA | Average |
|---|---|---|---|---|---|---|---|---|---|---|
| Full-FT | 8.5B | 72.32 | 87.32 | 76.86 | 91.07 | 81.76 | 92.46 | 82.76 | 89.00 | 84.19 |
| LoRA$_{r=32}$ | 68.8M | 71.55 | **87.95** | **77.27** | 91.80 | **79.71** | 92.67 | 82.16 | **86.40** | **83.69** |
| DoRA$_{r=16}$ | 35.5M | 71.46 | 87.59 | 76.35 | **92.11** | 78.29 | 92.00 | 80.63 | 85.60 | 83.00 |
| DoRA$_{r=1}$ | 3.31M | 68.22 | 86.72 | 75.23 | 91.14 | 78.13 | 91.87 | **83.19** | 86.20 | 82.59 |
| VeRA$_{r=2048}$ | 1.49M | 64.25 | 86.28 | 74.04 | 86.96 | 69.00 | 92.76 | 82.33 | 82.00 | 79.70 |
| LoRA$_{r=1}$ | 0.82M | 65.44 | 86.28 | 75.02 | 89.91 | 75.92 | 91.79 | 81.91 | 85.40 | 81.46 |
| SVFT$^P$ | 0.51M | 67.92 | 86.45 | 75.47 | 86.92 | 74.03 | 91.80 | 81.23 | 83.00 | 80.85 |
| SVFT$_{d=8}^B$ | 9.80M | **71.90** | 86.98 | 76.28 | 91.55 | 78.76 | **92.80** | 83.11 | 85.40 | 83.35 |

matches techniques with at least $3\times$ more trainable parameters. Notably, on GSM-8K, SVFT$_d^R$ again achieves **96%** of the full fine-tuning performance, while DoRA$_{r=16}$ recovers 86% with $2\times$ more parameters than SVFT$_d^R$.

**Commonsense Reasoning.** In Table 2, we compare performance on commonsense reasoning benchmarks with Gemma-7B, and observe similar trends. In the lower and moderately parameterized regime (∼0.43M), SVFT$^P$ shows competitive performance in comparison to LoRA$_{r=1}$ and DoRA$_{r=1}$, which have $1.9\times$ and $7.7\times$ more parameters, respectively. Against VeRA, which trains $3.5\times$ more parameters, SVFT$^P$ shows a relative improvement of ∼**1.16%**. Similarly, SVFT$_{d=8}^B$ also matches or exceeds methods that use up to $7\times$ more trainable parameters. For instance, SVFT$_{d=8}^B$ attains an average performance of 83.35% with only 9.8M parameters, closely matching LoRA$_{r=16}$ (83.69%, 68.8M parameters). We observe similar trends with Gemma-2B (refer Table 8).

**Natural Language Understanding.** Results on the GLUE benchmark are summarized in Table 3. SVFT

matches LoRA$_{r=8}$ and DoRA$_{r=4}$ which use **12-22×** more trainable parameters. Similarly, when compared to OFT and BOFT, SVFT$^P$ maintains a comparable average performance despite being $12\times$ smaller. These results highlight SVFT's ability to strike a balance between parameter efficiency and performance, making it an attractive PEFT choice for simple classification tasks.

**Parameter efficiency.** In Figure 1, we plot the performance of SVFT on mathematical reasoning and commonsense reasoning against other PEFT techniques across a range of configurations. Across trainable parameter budgets ranging from lowest to highest, SVFT obtains the best overall performance, matching methods that require significantly more trainable parameters. These results establish SVFT as a Pareto-dominant approach for parameter-efficient fine-tuning.

### 5.2. Performance on Vision Tasks

Table 4 contrasts SVFT against other PEFT techniques on image classification benchmarks using ViT-B and ViT-L models. For ViT-B, SVFT$_{d=8}^B$ surpasses full fine-tuning per-

*Table 3.* DeBERTaV3$_{base}$ with different adaptation methods on the GLUE benchmark. We report matched accuracy for MNLI, Matthew's correlation for CoLA, Pearson correlation for STS-B, and accuracy for other tasks. Higher is better for all tasks. * indicates numbers published in prior work.

| Method | #Params | MNLI | SST-2 | MRPC | CoLA | QNLI | QQP | RTE | STS-B | Avg. |
|---|---|---|---|---|---|---|---|---|---|---|
| Full-FT* | 184M | 89.90 | 95.63 | 89.46 | 69.19 | 94.03 | **92.40** | 83.75 | 91.60 | 88.25 |
| LoRA*$_{r=8}$ | 1.33M | **90.65** | 94.95 | 89.95 | 69.82 | 93.87 | 91.99 | 85.20 | 91.60 | 88.50 |
| DoRA$_{r=4}$ | 0.75M | 89.92 | 95.41 | 89.10 | 69.37 | 94.14 | 91.53 | 87.00 | 91.80 | 88.53 |
| BOFT*$_{m=2}^{b=8}$ | 0.75M | 90.25 | **96.44** | **92.40** | **72.95** | 94.23 | 92.10 | **88.81** | **91.92** | **89.89** |
| LoRA$_{r=1}$ | 0.17M | 90.12 | 95.64 | 86.43 | 69.13 | 94.18 | 91.43 | 87.36 | 91.52 | 88.23 |
| VeRA$_{r=1024}$ | 0.09M | 89.93 | 95.53 | 87.94 | 69.06 | 93.24 | 90.4 | 87.00 | 88.71 | 87.73 |
| SVFT$^P$ | 0.06M | 89.69 | 95.41 | 88.77 | 70.95 | **94.27** | 90.16 | 87.24 | 91.80 | 88.54 |
| SVFT$_{d=2}^R$ | 0.28M | 89.97 | 95.99 | 88.99 | 72.61 | 93.90 | 91.50 | 88.09 | 91.73 | 89.10 |

*Table 4.* Performance on image classification benchmarks. For LoRA, DoRA and SVFT$^B$, we adapt {Q, K, V, U, D} modules of the transformer. For SVFT$^P$, we adapt only {Q, V} to keep it comparable with VeRA. We report accuracy for all tasks.

| Method | ViT-B | | | ViT-L | | |
|---|---|---|---|---|---|---|
| | #Params | CIFAR100 | Flowers102 | #Params | Food101 | Resisc45 |
| Head | - | 78.25 | 98.42 | - | 75.57 | 64.10 |
| Full-FT | 85.8M | 85.35 | 98.37 | 303.3M | 77.83 | 76.83 |
| LoRA$_{r=8}$ | 1.32M | 84.10 | 99.23 | 3.54M | 77.13 | **79.62** |
| DoRA$_{r=8}$ | 1.41M | 85.03 | **99.30** | 3.76M | 76.41 | 78.32 |
| BOFT$_{m=4}^{b=4}$ | 0.11M | 85.54 | 98.59 | 2.95M | **78.42** | 74.70 |
| LoRA$_{r=1}$ | 0.16M | 84.86 | 96.88 | 0.44M | 75.97 | 78.02 |
| DoRA$_{r=1}$ | 0.25M | 84.46 | 99.15 | 0.66M | 75.90 | 78.02 |
| VeRA$_{r=256}$ | 24.6K | 83.38 | 98.59 | 0.06M | 75.97 | 72.44 |
| SVFT$^P$ | 18.5K | 83.85 | 98.93 | 0.05M | 75.95 | 71.97 |
| SVFT$_{d=2}^B$ | 0.27M | 84.72 | **99.28** | 0.74M | 77.94 | **79.70** |
| SVFT$_{d=8}^B$ | 0.93M | **85.69** | 98.88 | 2.5M | **78.36** | 73.83 |

formance along with LoRA$_{r=8}$ and DoRA$_{r=8}$ on CIFAR-100. SVFT$_{d=2}^B$ matches LoRA$_{r=8}$ and DoRA$_{r=8}$ on Flowers102 with up to $5\times$ fewer parameters. For ViT-L, SVFT$_d^B$ also demonstrates superior or competitive performance on both Food101 and Resisc45, with significantly lower trainable parameters compared to both fully fine-tuned models and other state-of-the-art PEFT approaches.

### 5.3. Contribution of Each Weight Type

In Figure 4, we investigate the contribution of each weight type. Starting with the base configuration, we apply SVFT$_d^B$ to the $Q$ and $V$ weights in each transformer block and report the performance. We then incrementally add the remaining weight modules ($K, U, D, O, G$) and observe the changes in performance. For each configuration, we also vary the trainable parameters by incrementing the total learnable off-diagonals.

Note that applying SVFT$_d^B$ to $U, D, O,$ and $G$ does not increase trainable parameters as much as applying LoRA/DoRA to these modules (Table 7). For example, for a large matrix of shape $d_1 \times d_2$, LoRA$_{r=1}$ learns $d_1 + d_2$ parameters, while SVFT$^P$ learns $\min(d_1, d_2)$ parameters. We observe that adapting only $U$ and $D$ with SVFT yields up to a 10% relative improvement over adapting $Q$ and $V$ for the same parameter budget ($\sim 0.8M$). Our findings indicate that adapting more weight types enhances performance.

### 5.4. Impact of $M$'s Structure on Performance

We analyze the impact of different parameterizations of $M$ (Plain, Banded, Random, Top-$k$) on downstream performance. To ensure a fair comparison, we maintain an equal number of trainable coefficients across all variants. As shown in Table 5, on Gemma-2B, both the Random and Top-$k$ variants outperform the Banded variant on the GSM-8K dataset. However, this improvement comes at the expense of performance on the MATH dataset. For larger models (Gemma-7B, LLaMA-3-8B), the Banded variant performs best on both tasks. These observations suggest that for smaller models, the choice of parameterization can significantly impact performance and may depend

*Table 5.* Results on fine-tuning with SVFT using different $M$ parameterizations.

| Structure | Gemma-2B | | | Gemma-7B | | | LLaMA-3-8B | | |
|---|---|---|---|---|---|---|---|---|---|
| | **#Params** | **GSM-8K** | **MATH** | **#Params** | **GSM-8K** | **MATH** | **#Params** | **GSM-8K** | **MATH** |
| Plain | 0.2M | 40.34 | 14.38 | 0.43M | 73.50 | 27.30 | 0.48M | 69.22 | 20.44 |
| Banded | 6.4M | 47.84 | 15.68 | 19.8M | 76.81 | 29.98 | 17.2M | 75.43 | 24.44 |
| Random | 6.4M | 50.03 | 15.56 | 19.8M | 76.35 | 29.86 | 17.2M | 74.07 | 23.78 |
| Top-$k$ | 6.4M | 49.65 | 15.32 | 19.8M | 76.34 | 29.72 | 17.2M | 73.69 | 23.96 |

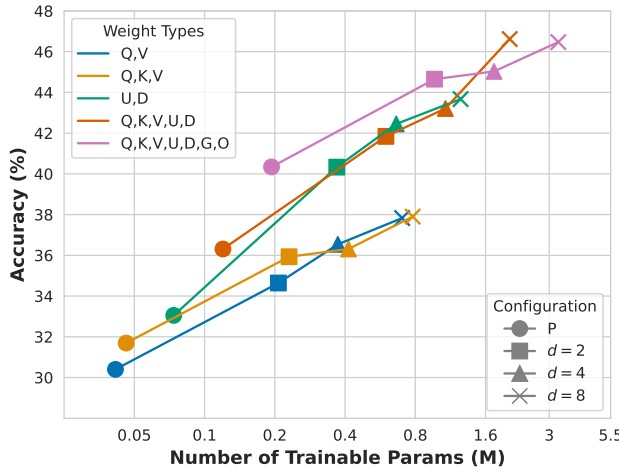

*Figure 4.* Performance variation with $\mathrm{SVFT}_d^B$ based on the adapted weight matrices – GSM-8K with Gemma-2B. Adapting more target weight types results in greater gains in performance. Interestingly, for a fixed parameter budget, adapting $U$ and $D$ weight types gives greater lifts than adapting $Q$ and $V$.

on the specific downstream task. In contrast, for larger models, Banded appears to be the better parameterization. Additionally, $\mathrm{SVFT}^P$ offers stronger performance as the model size increases.

*Table 6.* Impact of pre-trained weight quality. Results on GSM-8K after fine-tuning on Pythia-2.8B checkpoints at different stages of pre-training (PT). SVFT shows higher gains with better pre-trained weights. SVFT outperforms LoRA in both cases.

| Method | #Params | PT Steps | | $\Delta$**Perf** |
|---|---|---|---|---|
| | | **39K** | **143K** | |
| Full-FT | 2.5B | 21.00 | 30.09 | 9.09 |
| LoRA | 5.24M | 11.22 | 18.95 | 7.73 |
| SVFT | 5.56M | 15.08 | 23.19 | 8.11 |

### 5.5. Impact of Pre-trained Weight Quality

A key feature of SVFT is that the weight update depends on the pre-trained weights $W$. We therefore ask the following

question: *Does the quality of pre-trained weights have a disproportionate impact on* SVFT*?* To answer this, we consider two checkpoints from the Pythia suite (Biderman et al., 2023) at different stages of training, i.e., 39K steps and 143K steps, respectively. We fine-tune each of these checkpoints independently with Full-FT, LoRA, and SVFT. We then compare the increase in performance ($\Delta$Perf). As shown in Table 6, compared to LoRA, SVFT benefits more from better pre-trained weights. We also note that SVFT outperforms LoRA in both settings, suggesting that the benefits of inducing a $\Delta W$ that explicitly depends on $W$ are beneficial even when $W$ is sub-optimal.

## 6. Discussion

**Limitations.** Despite significantly reducing learnable parameters and boosting performance, SVFT incurs some additional GPU memory usage. Unlike LoRA and its variants, SVFT necessitates computing the SVD and storing both left and right singular vectors. While memory consumption remains lower than BOFT, it's roughly double that of LoRA. We mitigate this in our work by employing system-level optimizations like mixed-precision weights (e.g., bfloat16). However, similar to the scaling explored in (Wen & Chaudhuri, 2024), memory usage should amortize with the increasing scale of adaptation tasks. In future work we will explore quantization and other techniques to address memory concerns.

## 7. Conclusion

This work introduces SVFT, a novel and efficient PEFT approach that leverages the structure of pre-trained weights to determine weight update perturbations. We propose four simple yet effective sparse parameterization patterns, offering flexibility in controlling the model's expressivity and the number of learnable parameters. Extensive experiments on language and vision tasks demonstrate SVFT's effectiveness as a PEFT method across diverse parameter budgets. Furthermore, we theoretically show that SVFT can induce higher-rank perturbation updates compared to existing methods, for a fixed parameter budget. In future work, we aim to develop principled methods to generate sparsity patterns, potentially leading to further performance improvements.

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

# Appendix

The appendix is organized as follows.

- In Appendix A, we give proofs for the lemmas outlined in 3.2.

- In Appendix B, we compare the trainable parameter count for all baselines versus SVFT.

- In Appendix C, we describe results for additional experiments and provide implementation details for all the experiments.

## A. Proofs

We provide brief proofs for the *Structure*, *Expressivity* and the *Rank* lemmas for SVFT:

*(a) Structure:* If $M$ is diagonal, then the final matrix $W_0 + UMV^T$ can be written as
$U(\Sigma + M)V^T$ since $W_0 = U\Sigma V^T$, where $(\Sigma + M)$ is also a diagonal matrix. Thus, $U(\Sigma + M)V^T$ is a valid and unique SVD of $W_0 + UMV^T$ up to sign flips in the singular vectors.

*(b) Expressivity:* Finding $M$ for any target matrix $P$ of size $d_1 \times d_2$ such that $P = W_0 + UMV^T$ is the same as finding $M$ for a new target matrix $P' = P - W_0$ such that $P' = UMV^T$. For a full SVD, the dimension of $M$ is $d_1 \times d_2$ and since the dimension of $P'$ is also $d_1 \times d_2$, $P' = UMV^T$ is a bijection and $M = U^T(P - W_0)V$ (since $U$ and $V$ are orthogonal).

*(c) Rank:* If $M$ has $k$ non-zero elements, then the rank of the update $UMV^T$ will be upper bounded by $k$ (since by Gaussian elimination, $k$ or less elements will remain, the best case being all $k$ elements in the diagonal). We also know that the rank is upper bounded by $\min\{d_1, d_2\}$, giving an achievable upper bound on the rank as $\min\{k, \min\{d_1, d_2\}\}$.

## B. Parameter Count Analysis

*Table 7.* Parameter count analysis. $L_{\text{tuned}}$, $D_{\text{model}}$, $r$, $k$ denote total layers being adapted, hidden dimension, rank, and additional off-diagonals respectively.

| Method | Trainable Parameter Count |
|---|---|
| LoRA | $2 \times L_{\text{tuned}} \times D_{\text{model}} \times r$ |
| DoRA | $L_{\text{tuned}} \times D_{\text{model}} \times (2r + 1)$ |
| VeRA | $L_{\text{tuned}} \times (D_{\text{model}} + r)$ |
| SVFT$^P$ | $L_{\text{tuned}} \times D_{\text{model}}$ |
| SVFT$^B_{d=k}$ | $L_{\text{tuned}} \times (D_{\text{model}} \times k + (D_{\text{model}} - k)(k + 1))$ |

## C. Additional Experiments and Implementation Details

All of our experiments are conducted on a Linux machine (Debian GNU) with the following specifications: 2xA100 80 GB, Intel Xeon CPU @ 2.20GHz with 12 cores, and 192 GB RAM. For all our experiments (including baseline experiments), we utilize hardware-level optimizations like mixed weight precision (e.g., bfloat16) whenever possible.

### C.1. Commonsense Reasoning Gemma-2B

We evaluate and compare SVFT variants against baseline PEFT methods on commonsense reasoning tasks with Gemma-2B model and tabulate results in Table 8.

### C.2. Additional Vision Experiments

For vision tasks, we compare the SVFT banded variants and SVFT plain with baseline PEFT methods on classification vision tasks using ViT-Base and ViT-Large models in Table 9.

*Table 8.* Results with Gemma-2B on eight commonsense reasoning benchmarks. We follow (Liu et al., 2024a) for hyperparameter configurations, and report accuracy for all tasks.

| Method | #Params | BOOLQ | PIQA | SIQA | HellaSwag | Winogrande | ARC-E | ARC-C | OBQA | Average |
|---|---|---|---|---|---|---|---|---|---|---|
| Full-FT | 2.5B | 63.57 | 74.1 | 65.86 | 70.00 | 61.95 | 75.36 | 59.72 | 69 | 67.45 |
| LoRA$_{r=32}$ | 26.2M | 63.11 | 73.44 | 63.20 | 47.79 | 52.95 | 74.78 | 57.16 | 67.00 | 62.43 |
| LoRA$_{r=16}$ | 13.5M | 62.87 | 73.93 | 65.34 | 53.16 | 55.51 | 76.43 | 59.55 | 68.4 | 64.40 |
| BOFT$_{m=2}^{b=8}$ | 1.22M | 59.23 | 63.65 | 47.90 | 29.93 | 50.35 | 59.04 | 42.66 | 41.00 | 49.22 |
| VeRA$_{r=2048}$ | 0.66M | 62.11 | 64.31 | 49.18 | 32.00 | 50.74 | 58.08 | 42.83 | 42.6 | 50.23 |
| LoRA$_{r=1}$ | 0.82M | 62.2 | 69.31 | 56.24 | 32.47 | 51.53 | 69.52 | 48.8 | 56.4 | 55.81 |
| DoRA$_{r=1}$ | 1.19M | 62.17 | 68.77 | 55.93 | 32.95 | 51.22 | 68.81 | 48.72 | 55.6 | 55.52 |
| SVFT$^{P}$ | 0.19M | 62.26 | 70.18 | 56.7 | 32.47 | 47.04 | 69.31 | 50.08 | 58.4 | 55.81 |
| SVFT$_{d=16}^{B}$ | 6.35M | 63.42 | 73.72 | 63.86 | 71.21 | 59.58 | 73.69 | 54.77 | 66.6 | 65.86 |

*Table 9.* Performance on image classification benchmarks. For LoRA, DoRA and SVFT$_{d}^{B}$, we adapt {Q, K, V, U, D} modules of the transformer. For SVFT$^{P}$, we adapt only {Q, V} to keep it comparable with VeRA. We report accuracy for all tasks.

| Method | ViT-B | | | | ViT-L | | | | |
|---|---|---|---|---|---|---|---|---|---|
| | #Params | CIFAR100 | Flowers102 | Food101 | Resisc45 | #Params | CIFAR100 | Flowers102 | Food101 | Resisc45 |
| Head | - | 78.25 | 98.42 | 74.93 | 59.95 | - | 82.95 | 98.75 | 75.57 | 64.10 |
| Full-FT | 85.8M | 85.35 | 98.37 | 76.32 | 68.03 | 303.3M | 86.56 | 97.87 | 77.83 | 76.83 |
| LoRA$_{r=8}$ | 1.32M | 84.41 | 99.23 | 76.02 | 76.86 | 0.35M | 86.00 | 97.93 | 77.13 | 79.62 |
| DoRA$_{r=8}$ | 1.41M | 85.03 | 99.30 | 75.88 | 76.95 | 3.76M | 83.55 | 98.00 | 76.41 | 78.32 |
| BOFT$_{m=2}^{b=2}$ | 0.07M | 85.55 | 98.54 | 76.06 | 67.70 | 0.20M | 87.84 | 97.95 | 77.90 | 73.97 |
| BOFT$_{m=4}^{b=4}$ | 0.11M | 85.54 | 98.59 | 76.51 | 69.44 | 0.30M | 87.72 | 97.95 | 78.42 | 74.70 |
| LoRA$_{r=1}$ | 0.16M | 84.86 | 96.88 | 73.35 | 76.33 | 0.44M | 85.97 | 98.28 | 75.97 | 78.02 |
| DoRA$_{r=1}$ | 0.25M | 84.46 | 99.15 | 74.80 | 77.06 | 0.66M | 84.06 | 98.11 | 75.90 | 78.02 |
| VeRA | 24.6K | 83.38 | 98.59 | 75.99 | 70.43 | 61.4K | 86.77 | 98.94 | 75.97 | 72.44 |
| SVFT$^{P}$ | 18.5K | 83.85 | 98.93 | 75.68 | 67.19 | 49.2K | 86.74 | 97.56 | 75.95 | 71.97 |
| SVFT$_{d=2}^{B}$ | 0.28M | 84.72 | 99.28 | 75.64 | 72.49 | 0.74M | 86.59 | 98.24 | 77.94 | 79.70 |
| SVFT$_{d=4}^{B}$ | 0.50M | 83.17 | 98.52 | 76.54 | 66.65 | 1.32M | 87.10 | 97.71 | 76.67 | 71.10 |
| SVFT$_{d=8}^{B}$ | 0.94M | 85.69 | 98.88 | 76.70 | 70.41 | 2.50M | 87.26 | 97.89 | 78.36 | 73.83 |

## C.3. Are All Singular Vectors Important?

To determine the importance of considering all singular vectors and singular values during fine-tuning, we reduce the rank of $U$ and $V$, and truncate $\Sigma$ and $M$ to an effective rank of $r$. If the original weight matrix $W \in \mathbb{R}^{m \times n}$, then after truncation, $U \in \mathbb{R}^{m \times r}, V \in \mathbb{R}^{n \times r}$. This truncation significantly reduces the number of trainable parameters, so we compensate by increasing the number of off-diagonal coefficients ($d$) in $M$.

Our results, with four different configurations of $r$ and $d$, are presented in Table 10. The findings show that a very low rank ($r = 128$) leads to poor performance, even when parameters are matched. A reasonably high rank of $r = 1536$, which is 75% of the full rank, still fails to match the performance of the full-rank variant that has $0.25\times$ the trainable parameters. This indicates that all singular vectors significantly contribute to the end task performance when fine-tuning with SVFT, and that important information is lost even when truncating sparingly.

*Table 10.* Performance with varying rank ($r$) and the off-diagonal elements ($d$) of $M$. When $r = 2048$, the update is full-rank.

| Rank ($r$) | Diags ($d$) | #Params | GSM-8K | MATH |
|---|---|---|---|---|
| 128 | 64 | 1.55M | 0.98 | 0.21 |
| 1536 | - | 0.15M | 16.37 | 3.64 |
| 1536 | 2 | 0.74M | 25.01 | 6.04 |
| 2048 | - | 0.19M | **40.34** | **14.38** |

## C.4. Performance vs Total Trainable Parameters

In addition to the experiments performed in Figure 1 for Gemma-2B on challenging natural language generation (NLG) tasks like GSM-8K and Commonsense Reasoning, we also plot the performance vs total trainable parameters for larger state-of-the-art models like Gemma-7B and LLaMA-3-8B on GSM-8K. Figure 5 further demonstrates SVFT's Pereto-dominance. On larger models, we observe that full-finetuning overfits, leading to sub-optimal performance in comparison to PEFT methods.

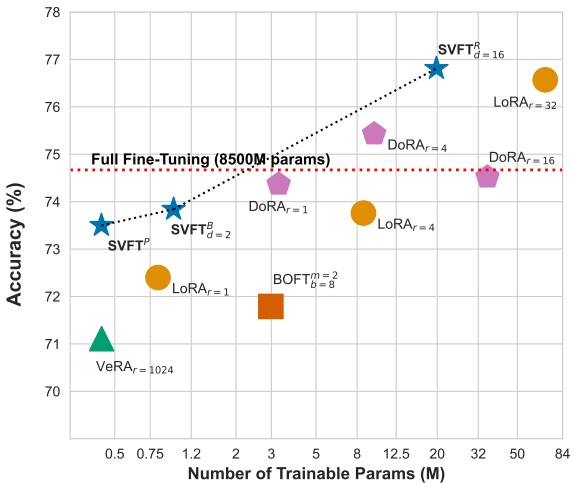 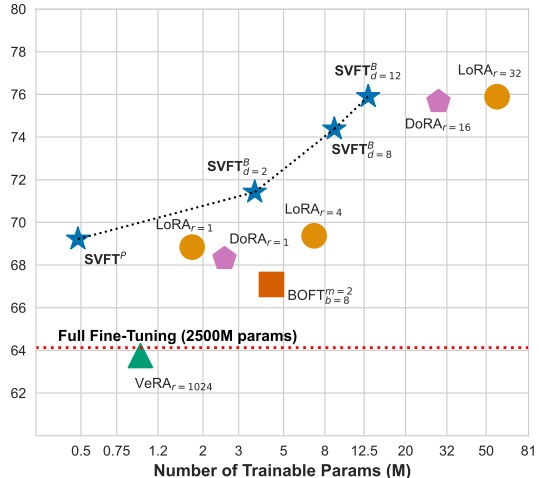

*Figure 5.* Performance versus total trainable parameters for GSM-8K on Gemma-7B (left) and LLaMA-3-8B (right).

## C.5. Settings for Language Tasks

**Natural Language Understanding.** We fine-tune the DeBERTaV3$_{base}$ (He et al., 2023) model and apply SVFT to all linear layers in every transformer block of the model. We only moderately tune the batch size, learning rate, and number of training epochs. We use the same model sequence lengths used by (Liu et al., 2024b) to keep our comparisons fair. The hyperparameters used in our experiments can be found in Table 11.

*Table 11.* Hyperparameter setup used for DeBERTaV3$_{base}$ on the GLUE benchmark.

| Method | Dataset | MNLI | SST-2 | MRPC | CoLA | QNLI | QQP | RTE | STS-B |
|---|---|---|---|---|---|---|---|---|---|
| | Optimizer | | | | AdamW | | | | |
| | Warmup Ratio | | | | 0.1 | | | | |
| | LR Schedule | | | | Linear | | | | |
| | Learning Rate (Head) | | | | 6E-03 | | | | |
| | Max Seq. Len. | 256 | 128 | 320 | 64 | 512 | 320 | 320 | 128 |
| | # Epochs | 10 | 10 | 30 | 20 | 10 | 6 | 15 | 15 |
| SVFT$^P$ | Batch Size | 32 | 32 | 16 | 16 | 32 | 16 | 4 | 32 |
| | Learning Rate | 5E-02 | 5E-02 | 5E-02 | 8E-02 | 8E-02 | 5E-02 | 5E-02 | 5E-02 |
| SVFT$^R_{d=2}$ | Batch Size | 32 | 32 | 16 | 16 | 32 | 32 | 16 | 32 |
| | Learning Rate | 1E-02 | 1E-02 | 1E-02 | 1E-02 | 3E-02 | 1E-02 | 3E-02 | 1E-02 |

**Natural Language Generation.** See the hyperparameters used in our experiments in Table 12. For LoRA, DoRA, we adapt $Q, K, V, U, D$ matrices. We apply BOFT on $Q, V$ matrices since applying on multiple modules is computationally expensive. For VeRA, which enforces a constraint of uniform internal dimensions for shared matrices, we apply on $G, U$ projection matrices as it yields the highest number of learnable parameters. We apply SVFT on $Q, K, V, U, D, O, G$ for the Gemma family of models, and $U, D, O, G$ for LLaMA-3-8B. Note that applying SVFT on these modules does not increase

trainable parameters at the same rate as applying LoRA or DoRA on them would. We adopt the code base from MetaMath[2] for training scripts and evaluation setups, and use the fine-tuning data available at MetaMathQA[3].

*Table 12.* Hyperparameter setup used for fine-tuning on MetaMathQA-40K.

| Hyperparameter | Gemma-2B | | Gemma-7B | | LLaMA-3-8B | |
|---|---|---|---|---|---|---|
| | $\text{SVFT}^P$ | $\text{SVFT}^R_{d=16}$ | $\text{SVFT}^P$ | $\text{SVFT}^R_{d=16}$ | $\text{SVFT}^P$ | $\text{SVFT}^R_{d=12}$ |
| Optimizer | | | AdamW | | | |
| Warmup Ratio | | | 0.1 | | | |
| LR Schedule | | | Cosine | | | |
| Learning Rate | 5E-02 | 1E-03 | 5E-02 | 1E-03 | 5E-02 | 1E-03 |
| Max Seq. Len. | | | 512 | | | |
| # Epochs | | | 2 | | | |
| Batch Size | | | 64 | | | |

**Commonsense Reasoning.** See the hyperparameters used in our experiments in Table 13. We adopt the same set of matrices as that of natural language generation tasks. We use the code base from https://github.com/AGI-Edgerunners/LLM-Adapters, which also contains the training and evaluation data.

*Table 13.* Hyperparameter setup used for fine-tuning on commonsense-15K.

| Hyperparameter | Gemma-2B | | Gemma-7B | |
|---|---|---|---|---|
| | $\text{SVFT}^P$ | $\text{SVFT}^B_{d=8}$ | $\text{SVFT}^P$ | $\text{SVFT}^B_{d=8}$ |
| Optimizer | | AdamW | | |
| Warmup Steps | | 100 | | |
| LR Schedule | | Linear | | |
| Max Seq. Len. | | 512 | | |
| # Epochs | | 3 | | |
| Batch Size | | 64 | | |
| Learning Rate | 5E-02 | 5E-03 | 5E-02 | 1E-03 |

*Table 14.* Hyperparameter setup used for fine-tuning on all vision tasks.

| Hyperparameter | ViT-B | ViT-L |
|---|---|---|
| Optimizer | | AdamW |
| Warmup Ratio | | 0.1 |
| Weight Decay | | 0.01 |
| LR Schedule | | Linear |
| # Epochs | | 10 |
| Batch Size | | 64 |
| $\text{SVFT}^P$ Learning Rate (Head) | | 4E-03 |
| $\text{SVFT}^P$ Learning Rate | | 5E-02 |
| $\text{SVFT}^B_{d=2}$ Learning Rate (Head) | | 4E-03 |
| $\text{SVFT}^B_{d=2}$ Learning Rate | | 5E-02 |
| $\text{SVFT}^B_{d=8}$ Learning Rate (Head) | | 4E-03 |
| $\text{SVFT}^B_{d=8}$ Learning Rate | | 5E-03 |

---

[2]https://github.com/meta-math/MetaMath.git
[3]https://huggingface.co/datasets/meta-math/MetaMathQA-40K

## C.6. Settings for Vision Tasks

For each dataset in the vision tasks, we train on 10 samples per class, using 2 examples per class for validation, and test on the full test set. Similar to previous literature, we always train the classifier head for these methods since the number of classes is large. The parameter counts do not include the number of parameters in the classification head. The hyperparameters are mentioned in Table 14. We tune the learning rates for SVFT and BOFT select learning rates for other methods from (Kopiczko et al., 2024), run training for 10 epochs, and report test accuracy for the best validation model. For all methods, since classification head has to be fully trained, we report the parameter count other than the classification head.

