# OpenReview forum: "SVFT: Parameter-Efficient Fine-Tuning with Singular Vectors"
_ICML.cc/2024/Workshop/WANT — WANT@ICML 2024 Oral_

### Official Review · Reviewer_2hmA · 2024-06-12
**Good paper, minor clarifications are required**

**Confidence:** 5

**Summary:**

This paper proposes a novel parameter-efficient LLM fine-tuning algorithm with focus on improving finetuning efficiency. The idea is to use frozen singular vectors of pre-trained weight matrices and a trainable scaling matrix to create a fine-tuned weight addition. The results indicate that the proposed method requires less number of parameters in general while maintaining or exceeding the performance of other similar PEFT approaches.

**Strengths:**

The paper is well-written very easy to read
The method is easy to understand
The experimental results are extensive.
All claims are backed up by experimental evidence.
Ablation experiments give further insights

**Weaknesses:**

I don't really see any weaknesses in this paper

**Limitations:**

- Singular vectors have to be stored for all pre-trained model weights, thus memory efficiency is noticeably worse than that of LoRA or other similar approaches.
- Method uses singular vectors of pre-trained weights, thus the quality of pre-train heavily influences the fine-tuning results, which is explicitly mentioned in the paper

**Suggestions:**

- In Appendix, section C.5 mentions using different groups of pre-trained models' weight matrices while comparing different PEFT approaches (e.g. BOFT - Q and K, VeRA - G and U). In my opinion it introduces a bias in PEFT performance comparison for natural language generation tasks.  I would like to see a cleaner comparison.
-  The choice of vision benchmarks is not fully clear to me. It would be interesting to see results on larger datasets and complex CV tasks such as image generation and image/video captioning.

---

### Official Review · Reviewer_Sr29 · 2024-06-13
**Novel idea for parameter efficient fine-tuning of Transformer models**

**Confidence:** 5

**Summary:**

This paper proposes a method to reduce the number of trainable parameters during fine-tuning, while improving the generalization. The proposed approach computes an SVD of the pre-trained weight, and initializes the adapter as reconstruction with SVD Eigen-vectors and Eigen-values. However, instead of learning all components, they only learn the correction to Eigen-values. I have seen this approach applied earlier in metric learning literature [1], but was nice to see it being revisited in this context. The experiments are quite exhaustive demonstrating the efficacy of their proposed approach.

**Strengths:**

- Method is clearly well motivated, and different variants for learning the Eigen values are presented. They also revisit related work and show how those solutions can be expressed as a special case of their generic formulation.
-  Really appreciate the ablations in Figure 4 (comparing various parameterizations) and study in section 5.5 analyzing the quality of pre-trained weights.
- Thorough evaluation on various tasks in NLP domain and CV domain.

**Weaknesses:**

- The only limitation I can think of is the extra memory overhead compared to LORA. This line of research was started to allow people fine-tune large GPT models on GPUs with limited memory. Authors have also noted this weakness in their paper. So, while the theoretical and empirical results are good, it would be great (potential future work) to mitigate this issue.

---

### Meta-Review · Area_Chair_jqen · 2024-06-17

**Recommendation:** Accept (Oral)
**Confidence:** 4

**Metareview:**

All reviewers champion the acceptance of this manuscript, given its well-motivated methodology, good paper writing quality, extensive empirical results, and thorough ablation study on various CV and NLP tasks.

---

### Decision · Program_Chairs · 2024-06-17

**Decision:**

Accept (Oral)

**Comment:**

We thank the authors for their time and contribution to WANT and we are pleased to share that after the reviewing process the paper has been accepted. Congratulations! We encourage the authors to consider reviewers' feedback for the improvement of the camera-ready version. We hope to see you in person at the workshop and brainstorm on efficient training research together!